# Intestinal Shedding of SARS-CoV-2 in Children: No Evidence for Infectious Potential

**DOI:** 10.3390/microorganisms11010033

**Published:** 2022-12-22

**Authors:** Filomena Nogueira, Klara Obrova, Meryl Haas, Evelyn Tucek, Karin Kosulin, Michaela Fortschegger, Paola Fürhacker, Christina Walter, Lisa Größlinger, Susanne Peter, Julia Othman Hassan, Martin Probst, Hans Salzer, Thomas Lion

**Affiliations:** 1Molecular Microbiology, St. Anna Children’s Cancer Research Institute, 1090 Vienna, Austria; 2Labdia Labordiagnostik GmbH, 1090 Vienna, Austria; 3St. Anna Children’s Hospital, 1090 Vienna, Austria; 4Department of Pediatrics, Universitätsklinikum Krems, 3500 Krems, Austria; 5Department of Pediatrics, Universitätsklinikum Tulln, 3430 Tulln, Austria; 6Department of Pediatrics, Medical University of Vienna, 1090 Vienna, Austria

**Keywords:** COVID-19, cytopathic effect, infectivity, pediatric, stool

## Abstract

The clinical courses of COVID-19 in children are often mild and may remain undiagnosed, but prolonged intestinal virus shedding has been documented, thus potentially enabling fecal–oral transmission. However, the infectious potential of SARS-CoV-2 viruses excreted with feces has remained unclear. Here, we investigated 247 stool specimens from 213 pediatric patients to assess the prevalence of intestinal SARS-CoV-2 shedding in hospitalized children without or with COVID-19 and determined the infectious capacity of stool-borne viruses. Upon RT-qPCR screening, the infectivity of virus-positive samples was tested in cell culture using the Vero-E6 permissive cell line. SARS-CoV-2 RNA was detected by RT-qPCR in 32 (13%) stool specimens, but the analysis of virus-positive samples in cell culture revealed no cytopathic effects attributable to SARS-CoV-2-related cell damage. Our findings do not support the notion of potential fecal–oral SARS-CoV-2 spreading, thus questioning the role of hygienic measures designed to prevent this mode of viral transmission.

## 1. Introduction

Regardless of the prevalent SARS-CoV-2 type [1], COVID-19 severe courses were less common in children, who can apparently cope much better with various viral diseases [2,3]. The COVID-19 hospitalization rates are associated with higher age [4] and the related exacerbation of inflammation with organ damage, which are mediated by activation of the immune system [5,6,7,8]. Children are more commonly asymptomatic or display very mild manifestations, and pediatric cases account for less than 1.5% of COVID-19-related hospitalizations [9]. Although underlying chronic disorders in children may be associated with an elevated risk for severe disease [10] and the occurrence of the Multisystem Inflammatory Syndrome (MIS-C) is a potentially serious complication associated with COVID-19 in a pediatric setting [11], current studies suggest that children are mostly able to mount an effective immune response with beneficial effects on inflammatory processes, generally showing better resolution of organ and tissue damage [12]. While children may be mostly protected against severe disease, there are open questions regarding their role as an important source for disease transmission.

In a pediatric study that included >2000 children, 4% had asymptomatic courses, 90% had mild or moderate clinical courses, and severe manifestations were documented in 6% of cases, mostly involving infants below 1 year of age [4]. Some reports indicated asymptomatic courses in as many as 16–30% of affected children within individual age groups [13,14]. In addition to common symptoms such as fever and dry cough (40–100% of cases), diarrhea was a frequent observation in children with COVID-19 (10–30% of cases), in contrast to adults with overt disease (4% of cases) [15,16,17,18,19,20]. Virus shedding into the stool was commonly observed in children for more than one month after resolution of acute infection [16,19,21]. However, the infectious potential of intestinal virus shedding, which has been demonstrated for the related viruses SARS-CoV and MERS-CoV, is currently unclear for SARS-CoV-2 [22]. Some studies have identified potential cell lysis by stool-borne SARS-CoV-2 [23,24,25,26,27,28], whereas others have not observed any cytopathic effect (CPE) [29,30]. In one of these studies, the failure to isolate viable SARS-CoV-2 from stools has been attributed to the mild disease observed in the investigated cohort. In this case, complete absence or very low levels of sgRNA (subgenomic RNA) indicative of viral replication were detected [30]. However, the indicated studies in pediatric settings were based on small patient cohorts. In case the virus remained viable upon intestinal shedding, stools might represent a source of infection and spread of SARS-CoV-2, with important implications for virus control measures and hygienic procedures. These would be relevant both in the hospital setting and in the daily lives of children and their caretakers, with particular importance for nurseries, kindergartens, and schools. If stool-borne SARS-CoV-2 remained infectious for prolonged periods of time, it could also be an indicator for viral persistence and could serve as an additional or alternative diagnostic tool. We hypothesized that a proportion of pediatric patients hospitalized for reasons unrelated to COVID-19 display intestinal shedding of SARS-CoV-2. Moreover, we intended to contribute to answering the question of whether stools of children positive for SARS-CoV-2 RNA may be infectious, regardless of the test results obtained from the upper respiratory tract.

## 2. Materials and Methods

### 2.1. Patients and Clinical Samples

Screening for SARS-CoV-2 was performed between December 2020 and June 2021 in 206 children hospitalized in three different clinical centers in Austria, i.e., the St. Anna Children’s Hospital in Vienna and the University Hospitals in Tulln and Krems, regardless of the indication for admission. The indications for hospitalization of children enrolled in the present study included mainly elective surgery, gastrointestinal symptoms, and various other problems mostly unrelated to acute infection. Screening of stool samples was also performed in seven children with upper respiratory tract infection who were not hospitalized at the time of investigation, three of whom were diagnosed with COVID-19. Overall, only seven children displayed overt COVID-19 (Table 1). Informed consent was obtained from each patient and/or their parents or legal representatives. The study was approved by the ethics committee of the University of Vienna (EK Nr. 1363/2020) and is registered at ClinicalTrials.gov (NCT05055466). The approval covered all three respective centers listed above.

For the analysis of fecal samples, native specimens were transferred to our laboratory in stool collection tubes at ambient temperature. Native patient stools were processed upon sample arrival for ensuing RNA extraction and RT-qPCR analysis for SARS-CoV-2 screening, and aliquots of positive samples were kept at 4 °C (which preserves virus infectivity better than dissolving in 0.9% NaCl and/or freezing at −80 °C) for further testing of viral viability in cell culture.

### 2.2. Cells

The Vero-E6 (ATCC CRL-1586) cell line, which is permissive for the propagation of SARS-CoV-2, was used in the present study. Cells were grown in 1× DMEM (Dulbecco’s Modified Eagle’s Medium, Gibco, Billings, MT, USA) + GlutaMax (Gibco, Billings, MT, USA) + 10% fetal bovine serum (FBS, Gibco, Billings, MT, USA) + 1× Penicillin/Streptomycin (P/S; Gibco, Billings, MT, USA) + 1× sodium pyruvate (SP) (Gibco, Billings, MT, USA) + 1× non-essential amino acids (NEAA; Sigma Aldrich, St. Louis, MO, USA), and were cultured at 37 °C in a 5% CO_2_ incubator.

### 2.3. RNA Extraction, cDNA Synthesis and Real Time Quantitative PCR (RT-qPCR)

Isolation of RNA was performed with the QIAamp Viral RNA extraction kit (Qiagen, Hilden, Germany) according to the manufacturer’s guidelines. For RT-qPCR, the primer/probe mix and controls from the Sarbeco e-gene kit (TIB MOLBIOL, Berlin, Germany) were used in combination with reagents from the Light Cycler Multiplex RNA Virus Master kit (Roche Diagnostics, Rotkreuz, Switzerland), and the reactions were performed in a LightCycler^®^ 480 instrument (Roche, Rotkreuz, Switzerland) according to the manufacturers’ guidelines. In addition to cell-based analyses of CPE (see details below), RNA extraction was performed from aliquots of cell culture medium taken on days 0, 4, and 7 post infection according to the guidelines described above. For cDNA synthesis, RNA aliquots were incubated at 70 °C for 5 min and placed on ice for 5 min. Master mix containing M-MLV RT reaction buffer, dNTPs, random hexamer primers, RNase inhibitor (RNasin) and M-MLV reverse transcriptase (all Promega, Madison, Wisconsin, United States) was added at a 1:1 volume ratio and the reaction was incubated at 37 °C for 60 min. For RT-qPCR, two different master mixes for the detection of SARS-CoV-2 genomic RNA (gRNA) and subgenomic RNA (sgRNA) were prepared using universal TaqMan™ PCR Master mix (Applied Biosystems, Waltham, Massachusetts, USA) and primer/probe sets targeting the E gene [30,31], encoding a structural protein as part of the viral envelope, were employed. The RT-qPCR reactions were performed on a 7500 TaqMan instrument (Applied Biosystems, Waltham, MA, USA) using the following amplification profile: 50 °C for 2 min, 95 °C for 10 min, and 45 cycles at 95 °C for 15 s, followed by 58 °C for 30 sec, with fluorescence signal acquisition as the last step in the cycle.

### 2.4. DNA Extraction and Real Time Quantitative PCR (RT-qPCR)

A subset of samples was tested for the presence of other viruses, with a particular focus on hAdVs. Isolation of DNA was performed with the DNA Minikit (Qiagen, Hilden, Germany) according to the manufacturer’s guidelines. Detection and quantification of viral DNA was performed using validated RT-qPCR according to protocols described previously [32,33,34,35].

### 2.5. Analysis of Virus Replication in Cell Culture

Prior to infection, the Vero E6 cells were trypsinized (trypsin from Gibco, Billings, MT, USA), mixed with 0.4% trypan blue dye (BioRad, Hercules, CA, USA) at a 1:1 ratio, and counted in an automated cell counter TC20 (BioRad, Hercules, CA, USA). Using 6-well plates, a total of 900 000 cells were seeded into each well containing 2 mL of DMEM + 10% FBS + P/S + SP + NEAA, and were then grown for 24 h at 37 °C prior to incubation with processed stool preparations.

In a biosafety level 2 (BSL2) environment, native stools (~1 cm^3^ aliquots kept at 4 °C) were resuspended in 1 mL DMEM 0 (without FBS) + P/S, spun down at 1600 rpm (558 g) for 5 min, and 200 µL of the supernatant were transferred into 400 µL DMEM 0 + P/S, sufficient for two technical replicates. Incubation of Vero-E6 cells with stool sample preparations was carried out in a BSL3 environment.

In two technical replicates, 500 µL of DMEM 0 and 300 µL of the stool suspension, were pipetted into each well and incubated for 1 h at 37 °C. The medium was then removed, the cells were washed with DMEM + 2% FBS + P/S + SP + NEAA and incubated in 2 mL of the same medium at 37 °C in a 5% CO_2_ incubator for a total of 7 days. The cells were inspected for potential cell lysis by light microscopy on a daily basis. At 0, 4, and 7 dpi, 140 µL aliquots of the medium were taken from each well and frozen at −80 °C for further RT-qPCR analysis.

The SARS-CoV-2 BU2-NS P3 strain (a kind gift of Dr. Jan Weber, IOCB, Prague, Czechia) was used as a positive control, while non-infected Vero-E6 cells (mock) served as a negative control. Controls were processed in the same way as described above.

## 3. Results

We investigated a total of 247 stool samples derived from 206 hospitalized pediatric patients with a median age of 4 years (range 0–17), comprising 48% females and 52% males. The patients had been admitted to the hospital for a variety of indications (see below), mostly unrelated to acute infection, to assess the prevalence of SARS-CoV-2 shedding into the stool in children without suspected or overt COVID-19. Only four children within this cohort had SARS-CoV-2 detectable in stool (Table 1). The samples studied also included stool specimens derived seven children with upper respiratory tract infection who were not hospitalized at the time of investigation, three of whom were diagnosed with COVID-19 (Table 1). Overall, RT-qPCR screening for SARS-CoV-2 revealed negative test results in 197 (79.8%) of the stool samples analyzed, while 32 (13%) specimens derived from seven patients tested positive for the virus, and 18 (7.3%) samples were not suitable for testing due to inadequate quality. All RT-qPCR positive stool samples were derived from patients with proven SARS-CoV-2 infection. To determine the viability of intestinally excreted SARS-CoV-2 in this setting, preparations from SARS-CoV-2 RNA-positive stool samples were processed and transferred onto permissive Vero-E6 cells under biosafety level 3 (BSL3) laboratory conditions. Subsequent screening for CPE was carried out over a period of seven days following inoculation. Inspection of the cell cultures by light microscopy was performed on a daily basis, and aliquots of the culture medium were taken at 0, 4, and 7 days post infection (dpi) for further analysis. SARS-CoV-2 positive stool specimens derived from seven patients were tested in cell culture, but a CPE was observed in two instances only (Table 1). The presence or absence of CPE did not correspond with the levels of SARS-CoV-2 RNA determined by RT-qPCR in the respective stool samples. To assess whether the CPE observed in two of the cell culture experiments performed was indeed related to SARS-CoV-2 replication in Vero-E6 cells, we conducted RT-qPCR analyses of the corresponding cultures to quantify the total or genomic RNA (gRNA) and subgenomic RNA (sgRNA) of SARS-CoV-2 at three sequential time points (0, 4 and 7 dpi). While control experiments using the viral strain SARS-CoV-2 BU2-NS P3 revealed elevated gRNA and sgRNA levels over time resulting from virus replication in Vero-E6 cells (see Appendix A), neither of the two patient samples displaying a CPE in Vero-E6 cells showed increased gRNA and sgRNA by RT-qPCR. These findings indicated that the CPE and the cell lysis observed in the indicated patient samples were apparently not attributable to SARS-CoV-2 replication. We have therefore screened these stool samples for the presence of other intestinally excreted viruses capable of infecting and lysing Vero-E6 cells in culture, including human Adenovirus species A-E (hAdVs), Cytomegalovirus and Enterovirus. Indeed, both stool samples contained hAdV species C, and one sample also contained Cytomegalovirus (CMV), albeit at low levels (AdV-C Ct values 36.64 and 35.24 for patient 4 and 5, respectively, CMV Ct value 36.1 for patient 5). We also identified hAdV species C in cell culture supernatants from both samples (Ct values 36.22 and 32.35 for patient 4 and 5, respectively), but CMV was not detectable in these specimens. Hence, it is conceivable that the CPE observed in these instances was attributable to a virus other than SARS-CoV-2.

## 4. Discussion

While some earlier reports based on anecdotal cases suggested the presence of infectious SARS-CoV-2 isolated from feces, other studies, also performed in small patient cohorts, were unable to identify any infectious potential of the virus excreted with stool [24,25,26,27,28,30]. In the present study, screening of stool specimens from more than 200 pediatric patients hospitalized for a variety of indications unrelated to COVID-19 revealed no intestinal SARS-CoV-2 shedding in this setting. Hence, the apparent absence of intestinal excretion of the virus in children without any clinical indication of respiratory infection with SARS-CoV-2 suggests that asymptomatic pediatric patients may not represent a relevant risk for SARS-CoV-2 spreading by the fecal–oral route, even if intestinally shed virus had infectious potential. Children revealing fecal excretion of SARS-CoV-2 RNA included exclusively individuals displaying documented respiratory infection with this virus. Previous studies have shown that mere detection of gRNA and sgRNA by RT-qPCR is not sufficient to identify the presence of infectious SARS-CoV-2 in diagnostic samples [36]. Although the levels of gRNA are expected to be higher compared to sgRNA, measuring sgRNA is specific for the assessment of actively replicating virus [30]. In the present study, we have therefore performed both cell-based assays and RT-qPCR analysis, including gRNA and sgRNA detection. However, cell-based in vitro analyses did not provide evidence for an infectious potential of the virus derived from stool in these patients, although all of them had overt COVID-19 at the time of analysis (Table 1). Vero-E6 cells, which were used for the cell culture experiments, are permissive for different viruses [37]. We therefore hypothesized that infection with another virus capable of infecting Vero-E6 cells could have caused the observed CPE. The corresponding samples were tested for the presence of select viruses, with a particular focus on the entire spectrum of hAdVs representing likely candidates in a pediatric setting [38,39,40]. We detected AdV species C in both stool samples and CMV at low levels in one of them. However, CMV displays a restricted host range and does not infect Vero-E6 cells [41], while Adenoviruses readily infect and lyse these cells [42,43,44]. Adenoviruses of species C were also detected in cell culture samples in both patients displaying CPE, suggesting that the cytopathic effect observed in Vero-E6 cells may have been attributable to these viruses.

The detectability and infectiousness of SARS-CoV-2 shed into the stool can be affected by various factors [45,46] including also enzymatic activity (e.g., RNAses), immune response, and stage or severity of the infection. In line with the latter notion, severe disease with high respiratory viral loads identified by RT-qPCR has been previously associated with a higher probability of successful virus propagation in cell culture and vice versa [47]. In patients shedding SARS-CoV-2 into the stool, the quantity of excreted virus was mostly rather low, as indicated by Ct levels of RT-qPCR analyses ranging from 22 to 38 (median 28). In our standardized assay, Ct values of 38 were established as the upper limit for reproducible detection of the virus, but due to the absence of harmonized diagnostic approaches, the reported thresholds of detectability vary between laboratories [48,49,50,51]. In some reports, Ct values above 30 were not considered as reliably positive, and accurate quantitative analyses were restricted by the dynamic ranges of individual assays [52]. Nevertheless, the number of SARS-CoV-2 positive stool specimens available for comprehensive testing, including virus propagation in cell culture, was a limitation also in the present study. Although it is conceivable that intestinal shedding of SARS-CoV-2 in patients with overt COVID-19 and high intestinal viral loads might also display a higher degree of infectious potential of stool-derived virus, it seems more relevant from the epidemiological perspective to identify the infectious capacity of viruses excreted into the stool by individuals not known or not suspected to be infected by SARS-CoV-2.

## 5. Conclusions

The observations made in the present study do not support the notion that the viruses are shed into the stool in a relevant proportion of children revealing no symptoms of COVID-19. Moreover, the absence of evidence for an infectious potential of intestinally shed SARS-CoV-2 in the investigated pediatric setting may suggest a low risk for fecal–oral transmission. The present findings therefore support the notion that special hygienic measures directed towards prevention of SARS-CoV-2 transmission via the fecal–oral route by children may be of limited epidemiological importance.

## Figures and Tables

**Table 1 microorganisms-11-00033-t001:** Cell culture test results in children with SARS-CoV-2 RNA positive stool samples. All patients displayed here were diagnosed with COVID-19. A selection of SARS-CoV-2 positive stool samples (*n* = 20) were tested in cell culture. CPE—cytopathic effect. * Hospitalized at the time of investigation.

Patient No.	Day(s) of Sample Collection	RT-qPCR Ct Value	CPE in Culture
1	0	27.4	No
	6	30.9	No
	11	29.6	No
	19	29.7	No
2	0	22.6	No
	2	26.0	No
	8	24.2	No
	14	24.5	No
	19	27.1	No
	25	35.3	No
	32	35.1	No
	39	37.4	No
3 *	0	28.7	No
	16	25.7	No
4	0	29.1	Yes
	12	32.8	No
	20	28.7	No
5 *	0	29.2	Yes
6 *	0	31.9	No
7 *	0	37.2	No

## Data Availability

The data presented in this study are available on request from the corresponding author.

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
