# Peer review of "Intestinal Shedding of SARS-CoV-2 in Children: No Evidence for Infectious Potential"

_microorganisms, 2022, doi:10.3390/microorganisms11010033_

Round 1

Reviewer 1 Report

The manuscript addresses a topic of interest for the scientific and public health worlds equally. Shedding of SARS-Cov-2 in the feces of infected people has been observed rather early in the pandemics but it was and still is a matter of uncertainty whether this is a means by which the virus can spread within a community. The study is not large but is well designed, conducted and described; it therefore contributes to understanding the issue and is recommended for publication after a few minor clarifications:

The samples have been collected from a number of centers and transported at ambient temperature; was there a time limit after which the samples were no longer suitable to be included in the study and if so, which was it?

It is stated that a number of viruses have been searched for as an alternative explanation for the CPE observed in a few cell cultures; please specify which were those viruses (only adenoviruses are specifically mentioned).

Please indicate what was the level of infectivity (Ct) in cell cultures for the reference material you have used in the experiments.

Is there available information about the level of viral presence and levels in respiratory samples of the positive patients? If so, please provide it.

Author Response

The samples have been collected from a number of centers and transported at ambient temperature; was there a time limit after which the samples were no longer suitable to be included in the study and if so, which was it?

We agree that a time limit can be helpful to ensure a uniform quality of incoming samples. However, such a limit was not set in the present study because our experience from routine diagnostics indicated that shipment of samples from centers in Austria generally arrive in our center in a timely manner permitting reliable and reproducible molecular analysis of viral infections. In general, the transport of samples for the current study was rather quick, permitting processing on the same day, and the great majority of specimens (n=232, 94%) could be processed within a maximum of 4 days post sampling– a time span that doesn´t pose any technical issue for molecular testing according to our long-standing diagnostic experience. In a few exceptional cases (n=4), transmission of samples was delayed for organizational reasons, but even specimens processed as late as 6 days post sampling still revealed detectable SARS-CoV-2 RNA.

It is stated that a number of viruses have been searched for as an alternative explanation for the CPE observed in a few cell cultures; please specify which were those viruses (only adenoviruses are specifically mentioned).

We tested for select stool-borne viruses commonly occurring in the clinical setting of the patient cohort involved displaying the capacity to lyse VeroE6 cells in culture, namely Cytomegalovirus, Adenovirus species A-E, and Enterovirus. This information has been added to the revised version of the manuscript (lines 187-188).

Please indicate what was the level of infectivity (Ct) in cell cultures for the reference material you have used in the experiments.

As correctly pointed out, the Ct values of reference virus have not been indicated in the text, but a fold change of viral RNA copies (calculated using a certified diagnostic assay) can be found in Figure S1. We provide the corresponding Ct values in the Table below for the Reviewer´s assessment.

Sample

Days post infection

Target

Ct1

Ct2

SARS-CoV-2 (lab strain)

0

gRNA

28.06

28.79

sgRNA

Undetermined

43.69

4

gRNA

18.53

18.61

sgRNA

30.86

31.08

7

gRNA

18.10

18.47

sgRNA

30.71

30.95

Mock

0

gRNA

Undetermined

Undetermined

sgRNA

Undetermined

Undetermined

4

gRNA

Undetermined

Undetermined

sgRNA

Undetermined

Undetermined

7

gRNA

Undetermined

Undetermined

sgRNA

Undetermined

Undetermined

Is there available information about the level of viral presence and levels in respiratory samples of the positive patients? If so, please provide it.

The seven patients, whose stool samples were positive for SARS-CoV-2 RNA, were all positive also in the respiratory tract and revealed clinical symptoms of COVID-19. The Ct values of positive pharyngeal swab samples were investigated at the clinical centers involved and ranged between 16.72 and 32.7. However, as individual qRT-PCR assays are not harmonized between the laboratories involved, these results cannot be compared in absolute terms. Indicating the numbers would therefore be of limited value and perhaps somewhat misleading.

Reviewer 2 Report

Comments on Nogueira et al:

The aim of this manuscript is to investigate and assess the prevalence of intestinal SARS-CoV-2 shedding, in hospitalized children, without or with COVID-19, and determined the infectious capacity of stool-borne viruses.

This manuscript shows rich content, providing a deep insight for some works: the study is within the journal’s scope, and I found it to be well-written, providing sufficient information. Even if the manuscript provides an organic overview, with a densely organized structure and based on well-synthetized evidence, there are some suggestions necessary to make the article complete and fully readable. For these reasons, the manuscript requires major changes.

Please find below an enumerated list of comments on my review of the manuscript:

INTRODUCTION:

LINE 32: The causative agent for COVID-19 is an enveloped positive single-stranded RNA virussevere acute respiratory syndrome coronavirus-2 (SARS-CoV-2)with the most prominent viral genome of 8.4–12 kDa in size. The viral genome includes a 5’ terminal in this viral genome, the central part of this genome, rich in open reading frames, which encodes proteins essential for virus replication. Instead, the 3’ terminal includes five structural proteins, Spike protein (S), membrane protein (M), nucleocapsid protein (N), envelope protein (E), and hemagglutinin-esterase protein (HE). This is the major concern of this manuscript: in this perspective, the authors should describe the main molecular features of SARS-CoV-2 viral genome, in order to provide sufficient evidence for expert and non -expert in the field (see, for reference: Torge, D.; Bernardi, S.; Arcangeli, M.; Bianchi, S. Histopathological Features of SARS-CoV-2 in Extrapulmonary Organ Infection: A Systematic Review of Literature. Pathogens 202211, 867. https://doi.org/10.3390/pathogens11080867).

LINE 34: The pediatric population is a vulnerable group, which requires a specific attention, during SARS-CoV-2 pandemics. In children, chronic underlying medical problems and a very young age are considered a significant risk factor, which may predispose to severe disease. Furthermore, the occurrence of asymptomatic and severe SARS-CoV-2 infection in children highlights the need to extend the vaccination also to the pediatric population (see, for reference: Khemiri, H., Ayouni, K., Triki, H., & Haddad-Boubaker, S. (2022). SARS-CoV-2 infection in pediatric population before and during the Delta (B. 1.617. 2) and Omicron (B. 1.1. 529) variants era. Virology journal19(1), 1-16).

The main topic is interesting, and certainly of great clinical impact. As regards the originality and strengths of this manuscript, this is a significant contribute to the ongoing research on this topic, as it extends the research field on the impact of intestinal SARS-CoV-2 shedding, in hospitalized children, without or with COVID-19Overall, the contents are rich, and the authors also give their deep insight for some works.

As regards the section of methods, there is a specific and detailed explanation for the methods used in this study: this is particularly significant, since the manuscript relies on a multitude of methodological and statistical analysis, to derive its conclusions. The methodology applied is overall correct, the results are reliable and adequately discussed.

The conclusion of this manuscript is perfectly in line with the main purpose of the paper: the authors have designed and conducted the study properly. As regards the conclusions, they are well written and present an adequate balance between the description of previous findings and the results presented by the authors.

Finally, this manuscript also shows a basic structure, properly divided and looks like very informative on this topic. Furthermore, figures and tables are complete, organized in an organic manner and easy to read.

In conclusion, this manuscript is densely presented and well organized, based on well-synthetized evidence. The authors were lucid in their style of writing, making it easy to read and understand the message, portrayed in the manuscript. Besides, the methodology design was appropriately implemented within the study. However, many of the topics are very concisely covered. This manuscript provided a comprehensive analysis of current knowledge in this field. Moreover, this research has futuristic importance and could be potential for future research. However, major concerns of this manuscript are with the introductive section: for these reasons, I have major comments for this section, for improvement before acceptance for publication. The article is accurate and provides relevant information on the topic and I have some major points to make, that may help to improve the quality of the current manuscript and maximize its scientific impact. I would accept this manuscript if the comments are addressed properly.

Author Response

LINE 32: The causative agent for COVID-19 is an enveloped positive single-stranded RNA virus, severe acute respiratory syndrome coronavirus-2 (SARS-CoV-2), with the most prominent viral genome of 8.4–12 kDa in size. The viral genome includes a 5’ terminal in this viral genome, the central part of this genome, rich in open reading frames, which encodes proteins essential for virus replication. Instead, the 3’ terminal includes five structural proteins, Spike protein (S), membrane protein (M), nucleocapsid protein (N), envelope protein (E), and hemagglutinin-esterase protein (HE). This is the major concern of this manuscript: in this perspective, the authors should describe the main molecular features of SARS-CoV-2 viral genome, in order to provide sufficient evidence for expert and non -expert in the field (see, for reference: Torge, D.; Bernardi, S.; Arcangeli, M.; Bianchi, S. Histopathological Features of SARS-CoV-2 in Extrapulmonary Organ Infection: A Systematic Review of Literature. Pathogens 202211, 867. https://doi.org/10.3390/pathogens11080867).

We agree that understanding the molecular features is an important prerequisite for understanding the current study, especially as RT-qPCR detection of two different targets has been used. Nevertheless, the genome structure of SARS-CoV-2 has been extensively described in the literature and has become common knowledge. We feel that addressing molecular features of the virus would be beyond the focus of the present study, and therefore chose to cite the relevant literature to guide the interested readers to the sources of appropriate information. The recent systematic review suggested by the Referee has been added to the reference list in the revised manuscript (line 32).

LINE 34: The pediatric population is a vulnerable group, which requires a specific attention, during SARS-CoV-2 pandemics. In children, chronic underlying medical problems and a very young age are considered a significant risk factor, which may predispose to severe disease. Furthermore, the occurrence of asymptomatic and severe SARS-CoV-2 infection in children highlights the need to extend the vaccination also to the pediatric population (see, for reference: Khemiri, H., Ayouni, K., Triki, H., & Haddad-Boubaker, S. (2022). SARS-CoV-2 infection in pediatric population before and during the Delta (B. 1.617. 2) and Omicron (B. 1.1. 529) variants era. Virology journal19(1), 1-16).

We agree that the pediatric population with chronic diseases represents a risk group for COVID-19 and is therefore an important target population for vaccination. Nevertheless, the current study was focused on the risk of fecal-oral spreading of the virus, and we feel that the certainly important aspect of vaccination in a specifically vulnerable pediatric population would exceed the scope of the present study. We have nevertheless included a pertinent statement in the Introduction of the revised manuscript (lines 38-39 and 41) and have also included the reference suggested by the Reviewer.

The main topic is interesting, and certainly of great clinical impact. As regards the originality and strengths of this manuscript, this is a significant contribute to the ongoing research on this topic, as it extends the research field on the impact of intestinal SARS-CoV-2 shedding, in hospitalized children, without or with COVID-19. Overall, the contents are rich, and the authors also give their deep insight for some works.

We greatly appreciate the positive assessment of the Reviewer.

Round 2

Reviewer 2 Report

Authors complied to the suggestions. Manuscript can be now accepted